# Evaluation of Satellite-Based Air Temperature Estimates at Eight Diverse Sites in Africa

**Danny Parsons [1],\*** , **David Stern [2]**, **Denis Ndanguza [1]** and **Mouhamadou Bamba Sylla [3]**

1. Department of Mathematics, School of Science, College of Science and Technology, University of Rwanda, Kigali P.O. Box 3900, Rwanda; dndanguzarusatsi@ur.ac.rw
2. IDEMS International, Reading RG2 7AX, UK; d.a.stern@idems.international
3. African Institute for Mathematical Sciences (AIMS), AIMS Rwanda Center, KN 3, Kigali P.O. Box 7150, Rwanda; sylla.bamba@aims.ac.rw
\* Correspondence: danny@aims.ac.za

**Abstract:** High resolution satellite and reanalysis-based air temperature estimates have huge potential to complement the sparse networks of air temperature measurements from ground stations in Africa. The recently released Climate Hazards Center Infrared Temperature with Stations (CHIRTS-daily) dataset provides daily minimum and maximum air temperature estimates on a near-global scale from 1983 to 2016. This study assesses the performance of CHIRTS-daily in comparison with measurements from eight ground stations in diverse locations across Africa from 1983 to 2016, benchmarked against the ERA5 and ERA5-Land reanalysis to understand its potential to provide localized temperature information. Compared to ERA5 and ERA5-Land, CHIRTS-daily maximum temperature has higher correlation and lower bias of daily, annual mean maximum and annual extreme maximum temperature. It also exhibits significant trends in annual mean maximum temperature, comparable to those from the station data. CHIRTS-daily minimum temperatures generally have higher correlation, but larger bias than ERA5 and ERA5-Land. However, the results indicate that CHIRTS-daily minimum temperature biases may be largely systematic and could potentially be corrected for. Overall, CHIRTS-daily is highly promising as it outperforms ERA5 and ERA5-Land in many areas, and exhibits good results across a small, but diverse set of sites in Africa. Further studies in specific geographic areas could help support these findings.

**Keywords:** climate; climate data; validation; temperature estimates; gridded data

## 1. Introduction

The availability of high-quality, long-term temperature records is important for a variety of applications that affect the lives and livelihoods of the population, as well as ecosystem services. This includes for health—in the study of heat waves [1–3] and the association between temperature and disease prevalence [4,5]; in agriculture—to understand the suitability of crops and their varieties under different and changing temperature patterns [6,7], including for crop simulation modeling [8,9]; and in understanding temperature increases caused by climate change [10,11], which are expected to affect Africa hardest [12,13].

In many cases, climate impacts are local [14,15], hence highly localized information is required. Historically, ground station measurements have been the primary source of air temperature data. However, Africa has the lowest density network of ground stations in the world [16]. Moreover, station networks are unevenly distributed and generally serve high population density areas [17], leaving many particularly rural areas lacking relevant local climate information. This presents a challenge to understand temperature patterns and hot extremes, thus preventing the provision of relevant strategies for adaptation and early warning at the local level.

The recent development of satellite- and reanalysis-based estimates of air temperature have the potential to complement the network of ground station measurements. They provide high-resolution, long-term records that could both enhance existing station records and provide temperature estimates in places where no historical measurements exist [18,19]. For instance, many crop simulation models require, among other elements, daily minimum and maximum air temperature to be provided with no missing values for the length of the study period [20,21]. These requirements can be extremely prohibitive to the adoption of such models across the continent if station data records are the only source of data. Even if a location of interest is near a ground station with temperature measurements, missing values that are often present in station records will prohibit the study. When located farther away from a station, there is a risk that results will not be applicable as climates can vary spatially, even over small distances. Long-term high-resolution satellite- or reanalysis-based temperature estimates could therefore enable studies that rely on complete temperature records to be more widely and more easily conducted.

One such product for temperatures is the recently developed Climate Hazards Center Infrared Temperature with Stations (CHIRTS-daily) dataset [18], which provides daily minimum and maximum air temperature estimates on a near-global scale from 1983 to 2016. CHIRTS-daily combines satellite infrared data with a large network of ground stations, as well as air temperature estimates from the ERA5 reanalysis from the European Centre for Medium-Range Weather Forecasts Re-Analysis (ECMWF) [19]. CHIRTS-daily is a particularly promising product for the applications mentioned because of its high resolution, which provides estimates approximately every 5 km (0.05° resolution). It has a higher resolution than other air temperatures estimates such as the Japanese 55-year reanalysis (0.5°) [22], ERA5 reanalysis (0.25°) and ERA5-Land reanalysis (0.1°) [23], and hence offers greater potential for providing localized temperature information.

The aim of this study is to evaluate the performance of CHIRTS-daily minimum and maximum temperature records at diverse locations in Africa through comparison with station data records from a selection of eight locations across Africa using a number of metrics. To benchmark the performance of CHIRTS-daily, two other high-resolution temperature products, ERA5 and ERA5-Land, are also compared to the station data records. The temperature estimates are evaluated on a daily basis, including some daily analysis split by month to account for seasonal variation and by year, where annual mean of minimum and maximum temperature, temperature extremes and trends are compared. We focus on a small number of stations in diverse locations across Africa where high-quality temperature records are available for 30 years or more.

While the sparsity of the station locations included in this study limits the confidence with which claims about specific locations can be made, the results will give an indication of performance in diverse climates across Africa and allow for evaluation of aspects of performance that require long-term records, such as temperature trends and annual means and extremes.

The rest of the paper is organized as follows. Section 2 describes the data and methods. Section 3 discusses the results. Conclusions, summary remarks and suggestions on future related research areas are presented in Section 4.

## 2. Materials and Methods

### 2.1. Study Sites

Eight sites across Africa are used in this study (see Figure 1). Although the choice of locations was ultimately restricted by the availability of long-term, high-quality daily temperature records, the sites represent diverse regions and climates across Africa, as categorized by the Köppen climate classification [24]. Four sites are above the equator, with Sadore, Niger in the Sahel having a hot semi-arid climate and the three stations in Ghana having a tropical savanna climate. Two of these three are in the hotter and drier northern Ghana. Kisumu, Kenya is on the equator, experiencing a tropical rainforest climate. In the Southern Hemisphere, Dodoma, Tanzania and Livingston, Zambia have a hot semi-arid

climate, whereas Mpika, Zambia experiences a humid subtropical climate. Two of the sites border water bodies, with Kisumu, Kenya on the shores of Lake Victoria and Saltpond on the south coast of Ghana bordering the Gulf of Guinea of the Atlantic Ocean.

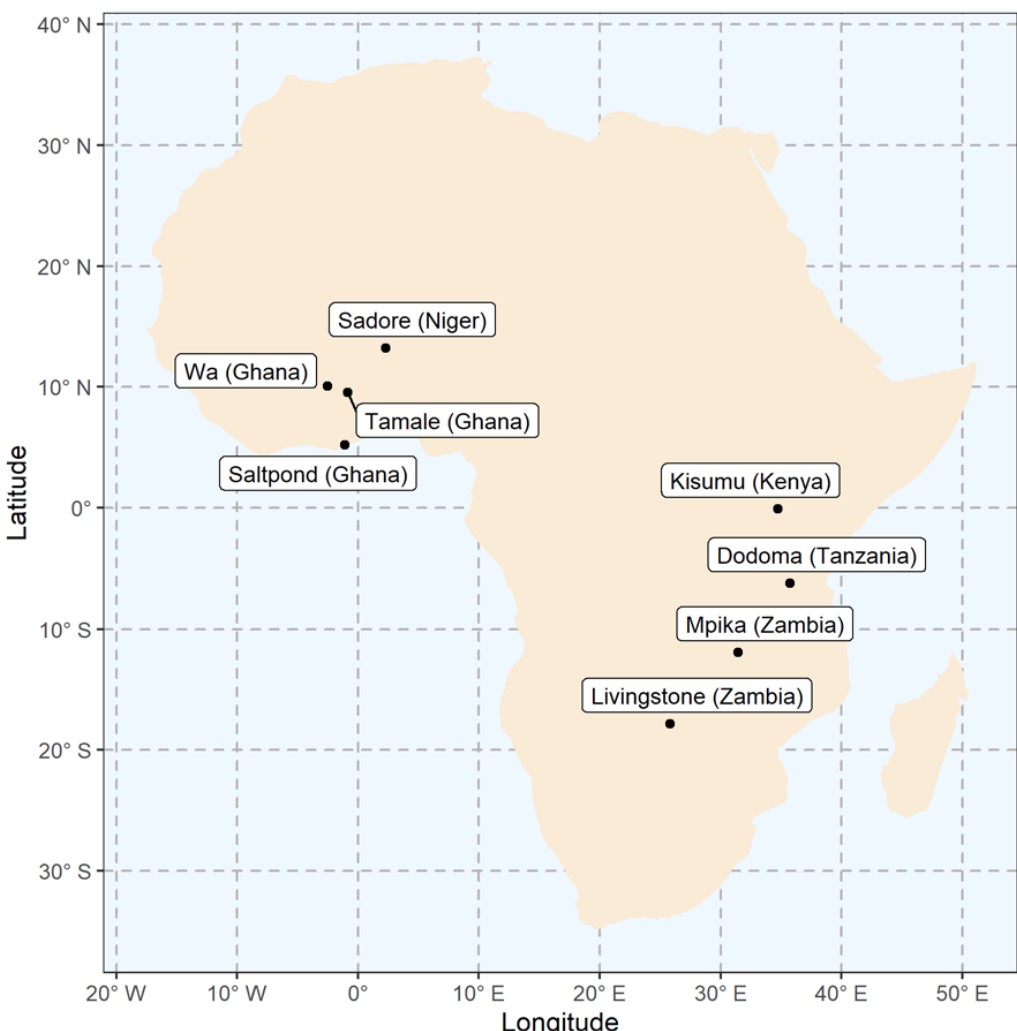

**Figure 1.** The location of the eight station sites.

### 2.2. Station Data

Daily minimum and maximum temperature station records were obtained at the eight sites. Station details and specifications about the temperature records are shown in Table 1. Daily rainfall was also obtained at the eight sites. The records from Ghana, Kenya, Tanzania and Zambia were obtained from the respective national meteorological services within those countries and the records from Sadore, Niger were obtained from the International Crops Research Institute for the Semi-Arid Tropics (ICRISAT).

Many of the station records begin much earlier than 1983 and some extend beyond 2016, however, only data within this period were used to coincide with the time period of CHIRTS-daily. The station records were quality controlled prior to analysis using a number of the consistency and statistical tests suggested by the World Meteorological Organization [25]. The tests used were: maximum and minimum temperature consistency (maximum > minimum), out-of-range check based on monthly climatological ranges, rapid change check of over 10 °C difference to the previous value and spike test with the same 10 °C difference threshold. The few values failing these quality-control checks were replaced by missing values. Graphical methods were also used to visually inspect the data

for further inconsistencies with no other issues found at any station. Most missing values in the station records were due to missing records from the data provider.

**Table 1.** Details of the eight station sites and properties of the station temperature data records.

| Country | Station | Latitude | Longitude | Data Range Used | Complete Days (%) |
|---------|---------|----------|-----------|-----------------|-------------------|
| Ghana | Wa | 10.05 | −2.5 | 1 January 1983–31 December 2012 | 97.1 |
| Ghana | Tamale | 9.55 | −0.86 | 1 January 1983–31 December 2016 | 93.3 |
| Ghana | Saltpond | 5.2 | −1.07 | 1 January 1983–31 December 2016 | 97.6 |
| Kenya | Kisumu | −0.08 | 34.73 | 1 January 1983–30 June 2014 | 100 |
| Niger | Sadore | 13.23 | 2.28 | 1 January 1983–10 September 2014 | 99.5 |
| Tanzania | Dodoma | −6.18 | 35.75 | 1 January 1983–31 October 2013 | 99.6 |
| Zambia | Mpika | −11.9 | 31.43 | 1 January 1983–30 April 2016 | 89.7 |
| Zambia | Livingstone | −17.82 | 25.82 | 1 January 1983–31 December 2016 | 90.8 |

### 2.3. Gridded Data

Temperature records from individual pixels of three gridded data products are compared to the station data records to investigate the added value from CHIRTS-daily. An overview of the key features of these products is given in Table 2.

The method to produce CHIRTS-daily begins by combining satellite infrared temperatures with a large collection of station temperature records from across the world to produce gridded monthly mean maximum temperatures, which are then combined with ERA5 temperature values to produce disaggregated daily maximum and minimum temperature values on a 0.05° quasi-global scale from 1983 to 2016: CHIRTS-daily [18], hereinafter referred to as CHIRTS.

The performance of CHIRTS is benchmarked against the performance of the 2 m temperature records from ERA5 and ERA5-Land reanalysis, developed by ECMWF. The ERA5 reanalysis combines global climate models with ground, ocean and satellite observations from a variety of sources using data assimilation systems to produce hourly estimates of a range of variables for the entire globe. ERA5 2 m temperature records are available as hourly instantaneous values from 1979 to the present (preliminary version from 1950) at a spatial resolution of 0.25° [19]. ERA5-Land is a downscaled version of ERA5's surface variables, aiming to provide greater detail over land with an improved 0.1° spatial resolution on the same hourly time scale [23].

Daily minimum and maximum temperature values were obtained from ERA5 and ERA5-Land by calculating the maximum and minimum of the 24 hourly values over each 24 h period starting at 6 AM UTC. ERA5-Land's inclusion adds a dataset with a closer spatial resolution to CHIRTS to reduce the likelihood that any improved performance of CHIRTS over ERA5/ERA5-Land can be solely attributed to its higher spatial resolution.

**Table 2.** Overview of gridded temperature products.

| Product | Spatial Resolution | Temporal Resolution | Data Availability | Coverage | Method |
|---------|--------------------|--------------------|--------------------|----------|--------|
| CHIRTS-daily | 0.05° (~5 km) | Daily | 1983–2016 | Quasi-Global | Merged Station, Satellite & Reanalysis |
| ERA5 | 0.25° (~30 km) | Hourly | 1979–Present | Global | Reanalysis |
| ERA5-Land | 0.1° (~9 km) | Hourly | 1981–Present | Global | Reanalysis |

### 2.4. Methodology

To compare the gridded data products with station measurements, the data from the closest grid point to each station were extracted from each of the gridded data products. This is often referred to as point-to-pixel comparisons [26–28]. The ERA5-Land grid point closest to the Saltpond station in Ghana partially contains sea and, hence, provides no values, so comparison at this site is not included for ERA5-Land. This is used to give an indication of performance at specific locations, rather than a large-scale spatial validation.

Comparison metrics were calculated from daily values and the annual mean of minimum and maximum temperatures. To analyze temperature extremes, the annual maximum temperature, i.e., the annual maximum of daily maximum temperature, was calculated. Potential seasonally dependent performance was accounted for by calculating comparison metrics of daily values on a monthly basis.

Comparison metrics used in this study were the Pearson's correlation coefficient, bias and root mean square error (RMSE), as defined in Equations (1)–(3), respectively.

$$\text{Bias} = \frac{1}{n} \sum_{i=1}^{n} e_i \tag{1}$$

$$\text{RMSE} = \sqrt{\frac{1}{n} \sum_{i=1}^{n} e_i^2} \tag{2}$$

$$\text{Correlation } (r) = \frac{\sum_{i=1}^{n} (s_i - \mu_s)(o_i - \mu_o)}{\sqrt{\sum_{i=1}^{n} (s_i - \mu_s)^2 \sum_{i=1}^{n} (o_i - \mu_o)^2}} \tag{3}$$

where $s_i$ is gridded data value at time $i$, $o_i$ is the station data value at time $i$, $e_i = s_i - o_i$ is the error at time $i$, $n$ is the length of the series and $\mu_s$ and $\mu_o$ are the means of gridded data and station data, respectively.

Following the suggestion in Knoben, Freer and Woods [29], which argues for "purpose-dependent evaluation metrics", the metrics chosen here allow for assessing multiple aspects of the performance of gridded products. Correlation shows the strength of agreement of values regardless of any biases. Bias detects systematic differences and its sign indicates long-term over- or underestimation. However, large deviations can be hidden by the bias as positive and negative deviations can offset each other. RMSE accounts for deviations and is therefore a measure of accuracy. We also calculate the standard deviation of the errors as another metric to indicate the consistency of any systematic bias. Taylor diagrams are used to summarize and compare performance in terms of pattern matching by simultaneously display correlation, centered root mean square error and standard deviation [30].

For the station data records, annual values were only calculated for years in which there were no more than 27 missing values and no more than 20 consecutive missing values. This condition is an adaptation of the World Meteorological Organization's recommendations for the calculation of monthly values for climate normals [31] and was chosen to prevent excessive missingness biasing the results, given the interannual variability of temperature at our locations and the autocorrelation of daily temperature values.

## 3. Results and Discussions

### 3.1. Station Data

Figure 2 shows the annual cycle of monthly mean minimum and maximum temperatures at each station, ordered by decreasing latitude. At the four stations in Niger and Ghana, maximum temperatures are regularly above 30 °C and highest in March to May, with Sadore, Niger having average maximum temperatures above 40 °C in April and May. At Kisumu, Kenya and Dodoma, Tanzania, maximum temperatures are fairly consistent year-round at 30 °C, whereas at the two stations in Zambia there is more seasonality with the highest temperatures in October and November.

Minimum temperatures exceed 20 °C throughout the year at the stations in Niger and Ghana. At Saltpond, the diurnal range is lowest at an average of less than 5 °C in July, whereas it is up to 15 °C at Sadore where minimum temperatures reach lower levels. At Kisumu, minimum temperatures are fairly consistent at around 17 °C. The farther south the stations go, the greater the minimum temperatures decrease in the winter months of July–September, with a slight seasonal variation in Dodoma, and greater temperature decreases at the stations in Zambia where average minimum temperatures are below 10 °C in this period. The proximity to the equator affects the seasonal patterns with stations closer to the equator (Saltpond, Kisumu and Dodoma) having less seasonal variability. The

stations farther from the equator exhibit a bimodal temperature pattern. In west Africa, temperatures reduce from June–September, because of the monsoon, and again in the winter months of December and January, with a hot dry season in-between that peaks in April. In Zambia, southern Africa, temperatures are highest in the hot dry season beginning in August, before reducing in October/November at the onset of the rainy season, which is then followed by a cool dry Southern Hemisphere winter. This highlights the diversity in seasonal patterns and temperature ranges of the sites included in this study.

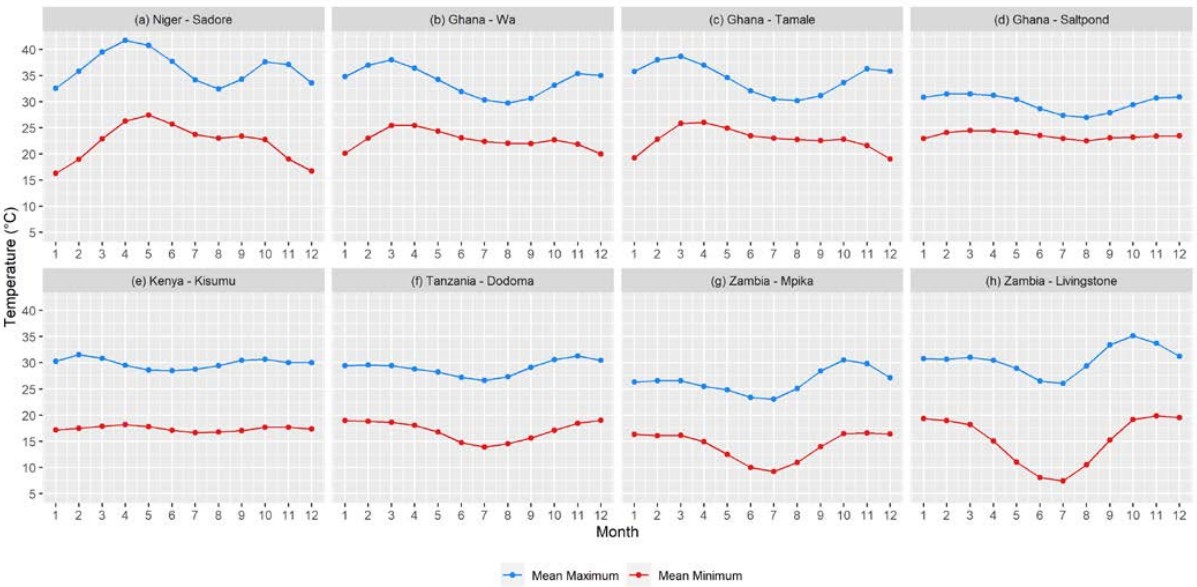

**Figure 2.** Annual cycle of monthly mean maximum (blue) and minimum (red) temperature from the stations in Niger and Ghana (**a–d**) and Kenya, Tanzania and Zambia (**e–h**).

### 3.2. Daily Comparisons

On a daily basis, the overall correlations between CHIRTS and the station data are high (r ≥ 0.80, not shown) for both minimum and maximum temperatures, and greater than or similar to those for ERA5 and ERA5-Land. However, the overall correlation can be misleading when there is seasonality in the data, as high correlation values could be achieved by values that model the seasonality well, without necessarily strong correlation of values within seasons. Hence, we also examine daily temperature correlations calculated by each month, i.e., 12 correlation values per site, alongside mean monthly rainfall from the station data, as shown in Figures 3 and 4.

CHIRTS has higher minimum temperature correlations than ERA5 and ERA5-Land in all or almost all months at each station. CHIRTS and ERA5 both have high correlations for maximum temperatures, with ERA5-Land having similar values for the stations in Kenya, Tanzania and Zambia and lower in those in Niger and Ghana. However, there is variation across the locations with Kisumu, and to a lesser extent Saltpond, having noticeably lower minimum temperature correlations than at the other stations for all three products. These results agree with the findings from the technical validation of CHIRTS in Verdin et al. [18], where the mean correlation with station data in Africa for the hottest three-month period was 0.81 and 0.67, respectively, for daily maximum and minimum temperatures.

We also note that ERA5-Land does not appear to have substantially higher correlations over ERA5 and actually has lower maximum temperature correlations at some stations. One would expect the improved downscaling to a higher spatial resolution to provide values that are more closely representative of point-based values.

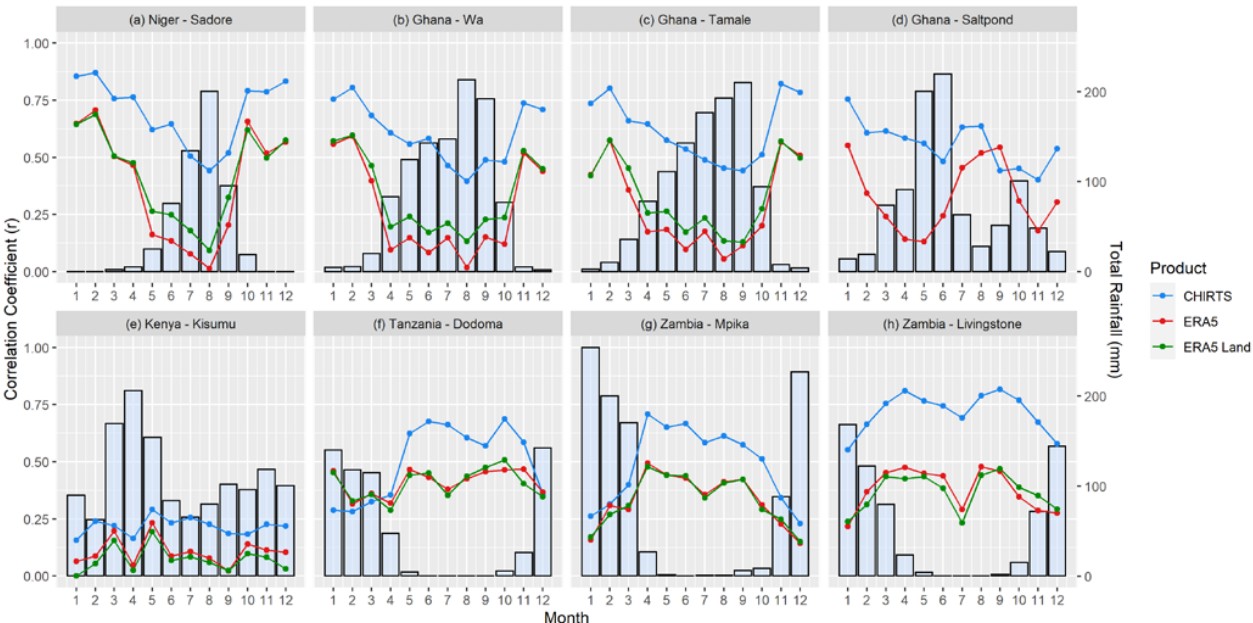

**Figure 3.** The lines show the correlation of daily minimum temperatures for each month between the station data and CHIRTS (blue), ERA5 (red) and ERA5-Land (green) at the stations in Niger and Ghana (**a**–**d**) and Kenya, Tanzania and Zambia (**e**–**h**). The light blue bars show the mean monthly total rainfall calculated from the station data.

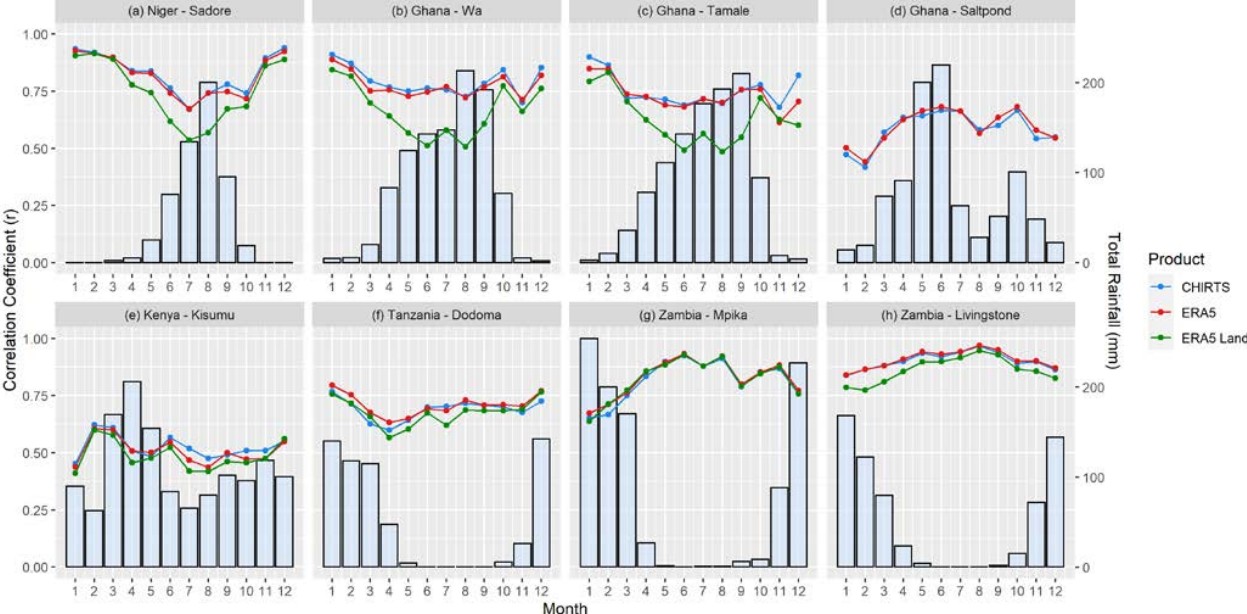

**Figure 4.** The lines show the correlation of daily maximum temperatures for each month between the station data and CHIRTS (blue), ERA5 (red) and ERA5-Land (green) at the stations in Niger and Ghana (**a**–**d**) and Kenya, Tanzania and Zambia (**e**–**h**). The light blue bars show the mean monthly total rainfall calculated from the station data.

There appears to be a seasonality effect in the daily minimum temperature correlations at some stations for all three products (Figure 3). At Sadore, Wa and Tamale, the minimum temperature correlations drop between June and September, corresponding to the main rainy season in these locations. Similarly, in the other locations, we generally also see a pattern of reduced correlations in the rainiest months. Temperature variability is also reduced during the rainy seasons, which can affect correlation. However, this does not correspond

to higher RMSE, which is fairly consistent across months, as shown in Figure 5. The lower correlation in these months is therefore likely due to lower variability of minimum temperature in this period since lower variability decreases correlation [32] and is not an indication of worse performance in terms of deviations in the rainy months as the consistent RMSE shows. For example, Figure 6 shows clearly that minimum temperatures at Sadore fall in a narrower range from July to September, where correlations are lower, even though the sizes of the differences are not larger than in other months. Hence, performance in estimating absolute values does not appear to be worse in those months. This also highlights the importance of using multiple metrics to assess different aspects of performance.

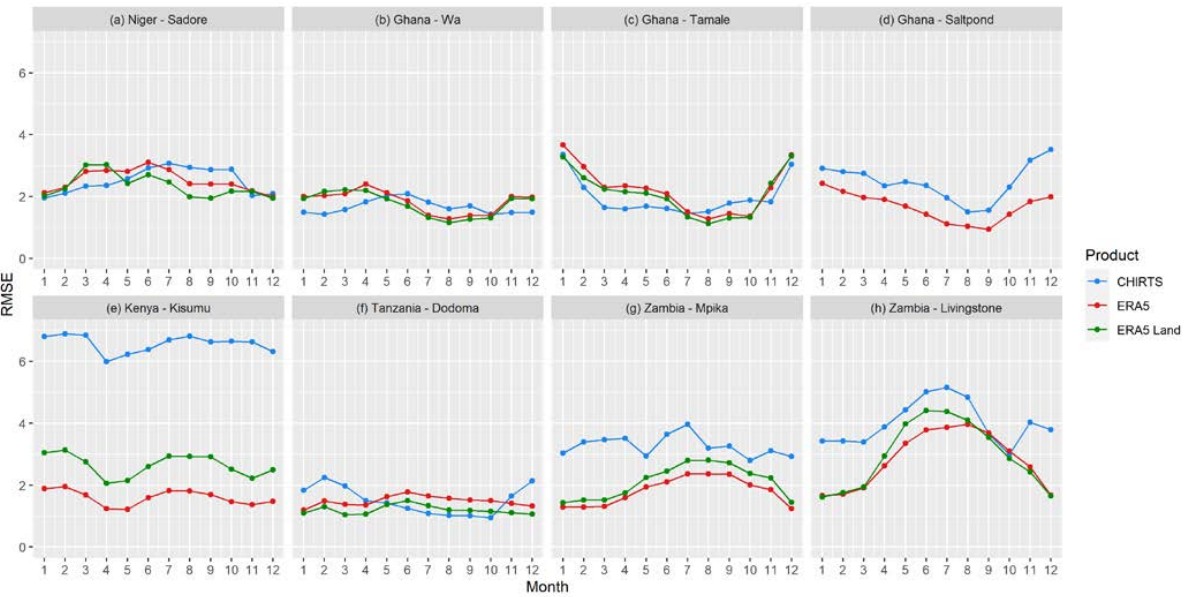

**Figure 5.** RMSE between station minimum temperatures and CHIRTS (blue), ERA5 (red) and ERA5-Land (green) for each month at the stations in Niger and Ghana (**a**–**d**) and Kenya, Tanzania and Zambia (**e**–**h**).

This effect is also observed to a lesser extent for maximum temperature correlations (Figure 4), although Dodoma, Tanzania and Saltpond, Ghana do not appear to follow this pattern.

CHIRTS overestimates daily minimum temperatures and by larger amounts on average than ERA5 and ERA5-Land at all stations (Table 3). The bias is largest at Kisumu (+6.4°). The daily mean bias is also large in Zambia at Mpika (+2.9°) and Livingstone (+3.7°), which has the lowest minimum temperatures of the eight sites. At the other stations, the overestimation is lower and between 0.6° and 2.3°. ERA5 and ERA5-Land also generally overestimate minimum temperatures but by less than CHIRTS at all stations. This is consistent with the RMSE values, which are generally similar or lower for ERA5 and ERA5-Land compared to CHIRTS. Wa is the only station where the minimum temperature is underestimated by CHIRTS for part of the year (November to March). CHIRTS consistently overestimates minimum temperatures in each month at all other stations (not shown); hence, only the overall bias values are reported. The larger bias in CHIRTS minimum temperatures compared to the corresponding bias for maximum temperatures was not observed in Verdin et al. [18], where mean absolute error over Africa was similar for minimum and maximum temperatures; however, these were only for the hottest three-month period in the year.

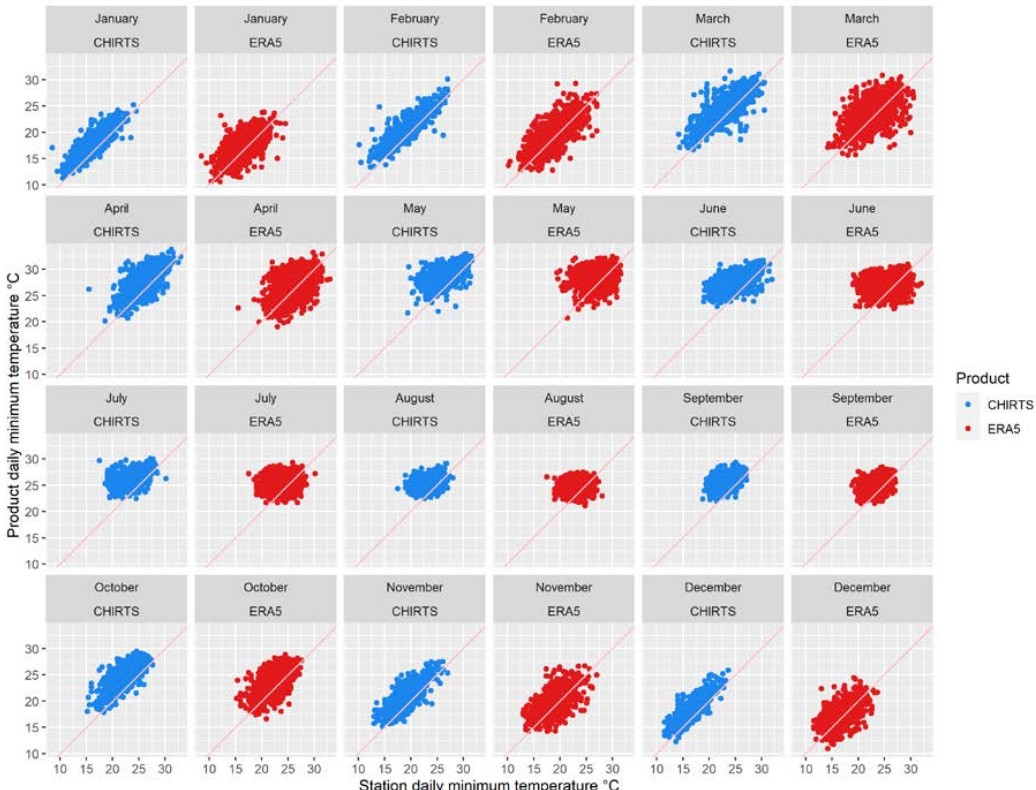

**Figure 6.** Scatter plots of station minimum temperature vs CHIRTS (blue) and ERA5 (red) for each month at Sadore, Niger.

**Table 3.** Bias (°C) between the three gridded products and the station data daily minimum and maximum temperatures at each of the eight locations.

| Location | Minimum Temperature | | | Maximum Temperature | | |
|---|---|---|---|---|---|---|
| | **CHIRTS** | **ERA5** | **ERA5-Land** | **CHIRTS** | **ERA5** | **ERA5-Land** |
| Sadore | 1.9 | 1.1 | 0.3 | 0 | −1 | −0.6 |
| Wa | 0.6 | 0 | −0.3 | −0.5 | −1.4 | −0.9 |
| Tamale | 1.2 | 1.0 | 0.7 | −0.6 | −1.1 | −0.7 |
| Saltpond | 2.3 | 1.2 | NA | 0.5 | −1.5 | NA |
| Kisumu | 6.4 | 0.6 | 2.2 | 1.4 | −4.4 | −3.5 |
| Dodoma | 1.0 | 0.9 | 0.2 | 0.2 | 0 | −0.6 |
| Mpika | 2.9 | 0.5 | 1.2 | 0.5 | −2 | −1.6 |
| Livingstone | 3.7 | 1.5 | 1.7 | 0.6 | −1.7 | −1.9 |

Although the CHIRTS minimum temperature biases are generally larger, the errors (differences) are slightly less variable with a lower standard deviation at all stations compared to ERA5 and ERA5-Land (Table 4). This is consistent with CHIRTS also having comparable or lower RMSE at some stations (Table 4), despite the higher biases. Therefore, the biases in the CHIRTS minimum temperatures may be largely systematic biases and, hence, could be corrected for. This is also consistent with the Taylor diagram in Figure 7, which shows smaller centered RMSE for CHIRTS since this considers the centered pattern error after subtracting the means. This result is illustrated in the scatter plots in Figure 7 of the station vs. the CHIRTS minimum temperatures at Sadore, Niger where the CHIRTS bias is almost double that of ERA5, yet the RMSEs are the same and CHIRTS has higher correlations. The CHIRTS data are more offset from the y = x line of perfect fit than the ERA5 data, but also appear to fit a straight line better; hence, with an offset correction CHIRTS values could be considered to better estimate the station data. From Figure 7 we

also observe that the standard deviations of all products are more often less than that of the station data, although not by large amounts, except at Kisumu. In general, the CHIRTS points are closest to the station point in Figure 7, showing better overall performance in these metrics.

**Table 4.** RMSE and standard deviation of error between the minimum temperatures of the station data and the gridded products.

| | RMSE | | | Standard Deviation of Error | | |
|---|---|---|---|---|---|---|
| Location | CHIRTS | ERA5 | ERA5-Land | CHIRTS | ERA5 | ERA5-Land |
| Sadore | 2.54 | 2.55 | 2.38 | 1.66 | 2.32 | 2.36 |
| Wa | 1.68 | 1.86 | 1.79 | 1.58 | 1.86 | 1.76 |
| Tamale | 2.06 | 2.37 | 2.22 | 1.65 | 2.15 | 2.11 |
| Saltpond | 2.54 | 1.72 | NA | 1.15 | 1.26 | NA |
| Kisumu | 6.57 | 1.62 | 2.67 | 1.43 | 1.51 | 1.52 |
| Dodoma | 1.56 | 1.49 | 1.21 | 1.19 | 1.19 | 1.19 |
| Mpika | 3.28 | 1.84 | 2.15 | 1.54 | 1.77 | 1.79 |
| Livingstone | 4.04 | 2.95 | 3.13 | 1.72 | 2.53 | 2.62 |

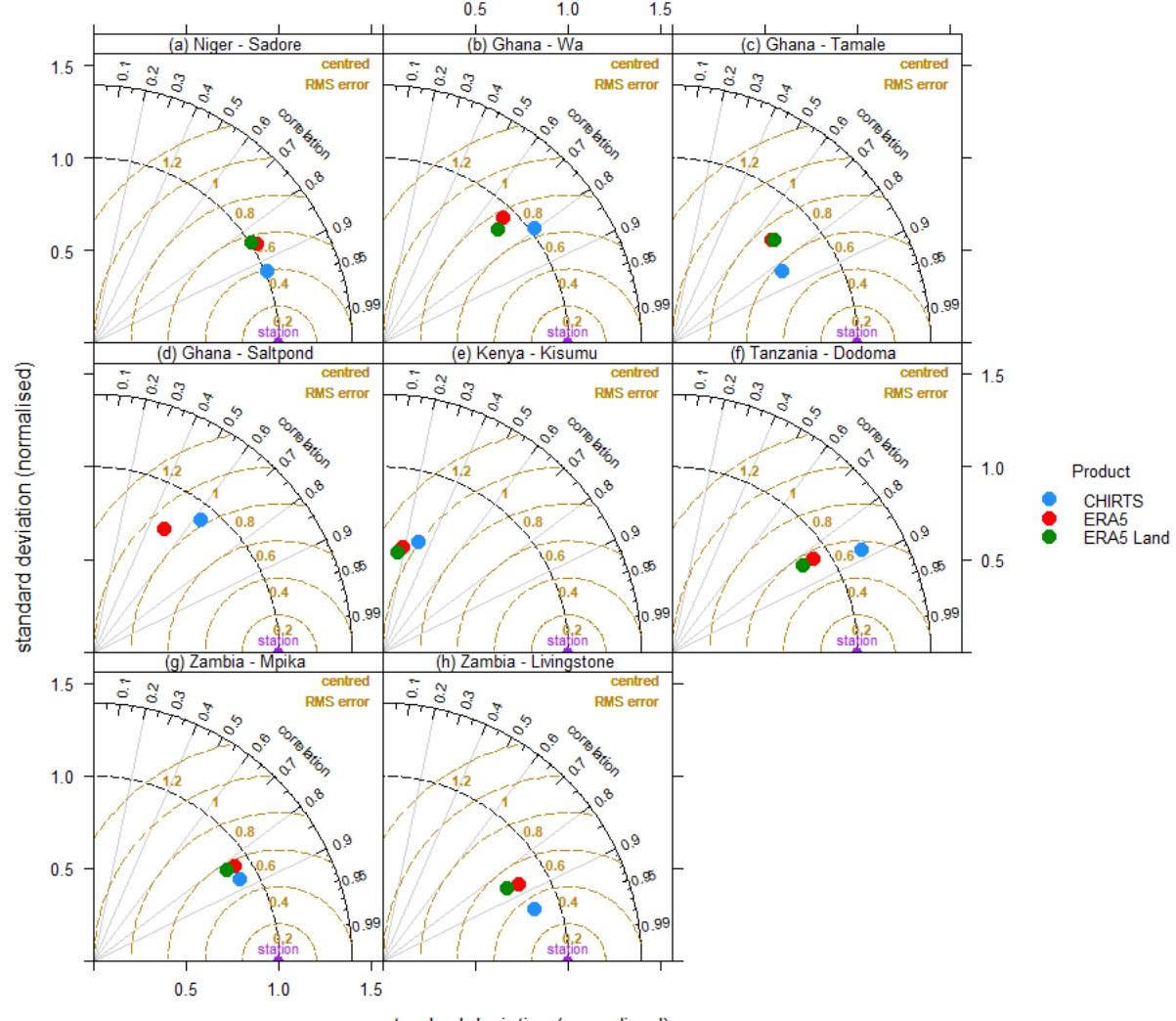

**Figure 7.** Taylor diagram comparing the performance of CHIRTS (blue), ERA5 (red) and ERA5 Land (green) with the station data (purple) by correlation, centered root mean square error and (normalized) standard deviation at the stations in Niger and Ghana (**a–d**) and Kenya, Tanzania and Zambia (**e–h**).

At all stations, CHIRTS has a lower bias for maximum temperature than it does for minimum temperature (Table 3). CHIRTS biases in maximum temperatures are consistently between −1.7° and 2° across all stations and months (Table 3). This range is even narrower if Kisumu is excluded. While CHIRTS consistently overestimates minimum temperatures, there is more of a mixture between under- and overestimation of maximum temperatures (Figure 8). CHIRTS overestimates in every month at Saltpond, Kisumu, Mpika and Livingstone, whereas it underestimates at Tamale and overestimates around the middle of the year and underestimates otherwise at Sadore and Wa (Figure 8).

ERA5 and ERA5-Land underestimate maximum temperatures on average and the biases are larger in size than for CHIRTS (Table 3). CHIRTS also has lower RMSE at all stations (Table 5). This improved performance of maximum temperature by CHIRTS over ERA5 and ERA5-Land is expected as the CHIRTS algorithm is designed to address the cool biases observed in ERA5 [33], which are more noticeable in Africa [18].

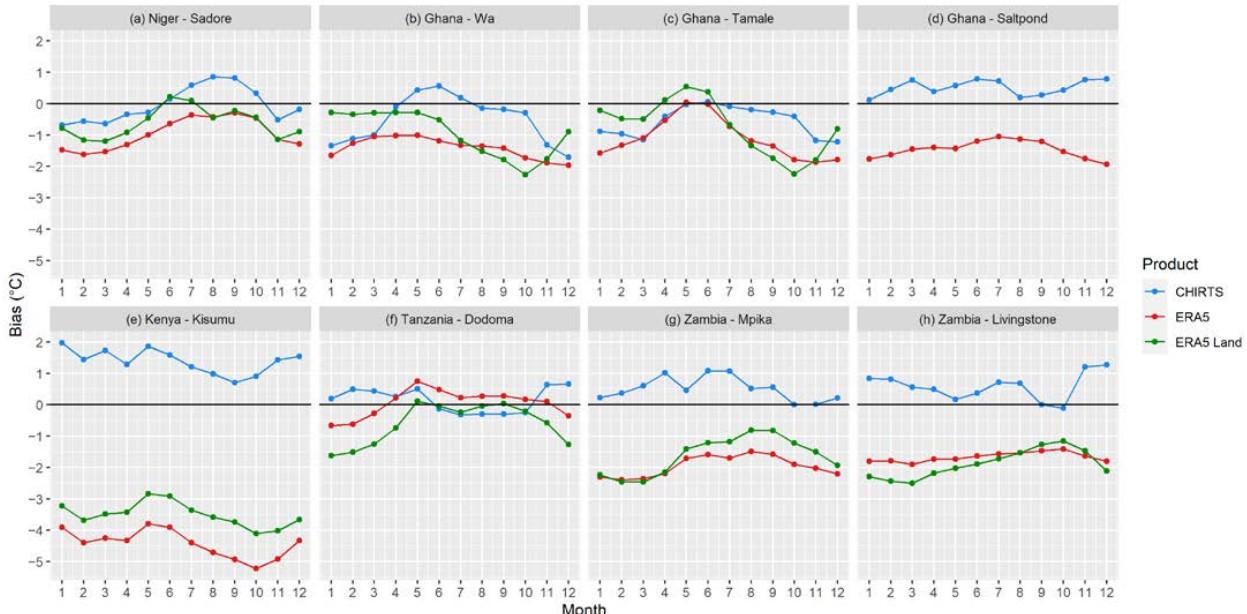

**Figure 8.** Bias between station maximum temperature and CHIRTS (blue), ERA5 (red) and ERA5-Land (green) for each month at the stations in Niger and Ghana (**a**–**d**) and Kenya, Tanzania and Zambia (**e**–**h**). The horizontal lines are at 0—a perfect bias score.

**Table 5.** RMSE between daily station maximum temperatures and the three gridded products.

| Location | CHIRTS | ERA5 | ERA5-Land |
|---|---|---|---|
| Sadore | 1.52 | 1.82 | 1.87 |
| Wa | 1.50 | 1.92 | 1.94 |
| Tamale | 1.48 | 1.85 | 1.94 |
| Saltpond | 1.05 | 1.69 | NA |
| Kisumu | 2.16 | 4.73 | 3.89 |
| Dodoma | 1.36 | 1.38 | 1.63 |
| Mpika | 1.41 | 2.34 | 2.14 |
| Livingstone | 1.43 | 2.06 | 2.42 |

Reda, Liu, Tang, and Gebremicael [34] also showed that CHIRTS exhibits good performance on a daily basis in comparison with records at stations in the complex terrain of the Upper Tekeze River Basin, Ethiopia and performed better than other products. However, it is noticeable that the average daily correlations were lower than observed in this study and the RMSE values were higher at 3.7° and 4° for maximum and minimum temperatures, respectively.

The performance of CHIRTS at Kisumu, Kenya stands out as being the poorest across all measures. This could relate to the complex local climate around Kisumu. It is on the shores of Lake Victoria, 24 km from the equator, and is close to much cooler, higher-altitude and mountainous areas. However, this would need further investigation in a more detailed study focused on this area with more station records. At Saltpond, Ghana, CHIRTS also has a relatively worse performance compared to its closest other stations, and Saltpond also borders a water body close to the equator. However, there were also some large biases and low correlations in some months in the two locations in Zambia, which does not have these features.

Although it was not an objective of the study, it is somewhat surprising that the higher resolution of ERA5-Land compared to ERA5 does not appear to improve performance in comparison to the station data in point-to-pixel comparisons. One would have expected temperature values to be dependent, for example, on the pixel altitude and, hence, a smaller pixel area around the station location may be expected to compare better to the station data. A further study to understand the spatial variability of ERA5 and ERA5-Land over Africa could help to better understand this.

### 3.3. Annual Means

The CHIRTS annual mean minimum temperatures are positively correlated with the station values. Kisumu and Dodoma have the lowest correlations values (0.26 and 0.32) (Table 6). ERA5 and ERA5-Land generally have similar or higher correlation with the station values. The larger systematic biases observed in the daily CHIRTS minimum temperature could be a cause for the relatively worse performance of the yearly means.

**Table 6.** Correlation between station and product mean minimum temperatures and station and product mean maximum temperatures.

| Location | Mean Minimum Temperature | | | Mean Maximum Temperature | | |
|---|---|---|---|---|---|---|
| | CHIRTS | ERA5 | ERA5-Land | CHIRTS | ERA5 | ERA5-Land |
| Sadore | 0.65 | 0.83 | 0.84 | 0.73 | 0.44 | 0.61 |
| Wa | 0.56 | 0.59 | 0.63 | 0.72 | 0.6 | 0.74 |
| Tamale | 0.82 | 0.74 | 0.77 | 0.56 | 0.47 | 0.74 |
| Saltpond | 0.66 | 0.86 | NA | 0.81 | 0.79 | NA |
| Kisumu | 0.26 | 0.44 | 0.63 | 0.73 | 0.65 | 0.63 |
| Dodoma | 0.32 | 0.86 | 0.8 | 0.87 | 0.8 | 0.81 |
| Mpika | 0.78 | 0.78 | 0.77 | 0.91 | 0.94 | 0.89 |
| Livingstone | 0.61 | 0.65 | 0.48 | 0.93 | 0.92 | 0.81 |

CHIRTS annual mean maximum temperatures are highly correlated with the station values and at each station the correlation is higher than the corresponding correlation for mean minimum temperature. Compared to ERA5 and ERA5-Land, CHIRTS has higher correlation, except at Tamale, where ERA5-Land has a higher correlation. The improved performance of CHIRTS annual mean maximum temperatures is consistent with the results of Verdin et al. [18] in the technical validation of the CHIRTS product.

The station data show trends of increasing annual mean maximum temperatures between 0.43 and 3.07 °C per 100 years, which are significant at the 5% level in all stations except Mpika (Table 7). CHIRTS shows increasing trends in mean maximum temperatures, which are significant at the 5% level at five of these six locations, compared to three out of six for ERA5. The significant trends in CHIRTS are fairly consistent, between 1.26 and 2.77 °C per 100 years, and similar to those of the station data.

**Table 7.** Annual mean maximum temperature trend per 100 years (°C) and corresponding *p* value for station data and the three gridded products.

| | Station | | | CHIRTS | | ERA5 | | ERA5-Land | |
|---|---|---|---|---|---|---|---|---|---|
| **Station** | **Trend** | **p Value** | **Trend** | **p Value** | **Trend** | **p Value** | **Trend** | **p Value** |
| Sadore | 2.81 | 0.004 | 2.77 | <0.001 | 1.39 | 0.097 | 0.83 | 0.327 |
| Wa | 2.56 | 0.002 | 1.75 | 0.005 | 2.02 | 0.072 | 1.7 | 0.059 |
| Tamale | 0.43 | 0.599 | 1.65 | <0.001 | 1.16 | 0.176 | 1.68 | 0.017 |
| Saltpond | 3.07 | <0.001 | 1.86 | 0.002 | 2.19 | <0.001 | NA | NA |
| Kisumu | 1.18 | 0.085 | 1.98 | <0.001 | 3.7 | <0.001 | 3.75 | <0.001 |
| Dodoma | 1.57 | 0.039 | 1.26 | 0.027 | 3.57 | <0.001 | 2.8 | 0.002 |
| Mpika | 1.07 | 0.257 | 1.12 | 0.12 | 1.2 | 0.121 | 0.69 | 0.394 |
| Livingstone | 0.9 | 0.457 | 1.02 | 0.337 | 1.38 | 0.227 | −1.14 | 0.416 |

*3.4. Extremes*

Compared to ERA5 and ERA5-Land, CHIRTS annual extreme maximum temperatures are closer to the station values on average (Table 8). CHIRTS underestimates the average annual extreme maximum temperatures in the three most-northern stations, which have the highest extremes, by between 0.53 and 0.8 °C. At Saltpond and Mpika, the CHIRTS bias is close to 0. At the other three stations, CHIRTS overestimates by 0.51 to 1.01 °C on average. ERA5 and ERA5-Land generally underestimates the extreme maximum temperatures and mostly by a larger magnitude than the CHIRTS bias. This is consistent with one of the aims of CHIRTS to improve the cool bias of ERA5 and specifically its underestimation of extremely hot days [18] and is also consistent with Reda, Liu, Tang and Gebremicael [34], who found that maximum temperature extremes were better represented by CHIRTS than other products over the Upper Tekeze River Basin, Ethiopia.

**Table 8.** Bias between the annual extreme maximum temperature from the station data and the three gridded products.

| Location | CHIRTS | ERA5 | ERA5-Land |
|---|---|---|---|
| Sadore | −0.7 | −1.48 | −1.45 |
| Wa | −0.53 | −0.89 | −0.6 |
| Tamale | −0.8 | −0.93 | −0.69 |
| Saltpond | −0.06 | −2.52 | NA |
| Kisumu | 0.92 | −5.08 | −4.23 |
| Dodoma | 1.01 | 0.19 | −0.57 |
| Mpika | −0.12 | −2.26 | −1.95 |
| Livingstone | 0.51 | −1.67 | −1.39 |

**4. Conclusions**

This study presents results comparing temperature data records of 30 years or more from eight stations in Africa with the CHIRTS satellite- and station-based daily temperature records from 1983 to 2016. The station data were also compared to ERA5 and ERA5-Land temperatures to benchmark the CHIRTS performance. The study focused on the analysis of a small number of diverse locations in detail, as opposed to a larger number of locations in a dense area. While this approach limits the ability to generalize these results across the continent, it does provide an indication of the performance of CHIRTS in a variety of climates. The results for maximum temperature are extremely promising across measures and locations. The minimum temperature results are more complex, but still promising in some key measures.

The CHIRTS daily minimum temperatures generally have higher correlation with the station data than ERA5 and ERA5-Land.

CHIRTS overestimates minimum temperatures on average across the locations, with daily bias above 2 °C at four of the eight stations and highest at Kisumu, Kenya at 6.4 °C. ERA5 and ERA5-Land also overestimated generally but, in contrast, the ERA5 daily bias

was below 1.5 °C at all stations. The larger CHIRTS biases could be an effect of the CHIRTS algorithm. CHIRTS minimum temperatures are calculated by subtracting the ERA5 diurnal range from the CHIRTS maximum temperatures [18]. Since we have also observed that CHIRTS maximum temperatures are generally increased compared to ERA5 values to account for ERA5 cool bias, a possible explanation is that this method may have inadvertently led to increased minimum temperature overestimation. For many applications where minimum temperature values are directly required (as opposed to relative values), such as in the study of tropical disease transmission and epidemics [35,36] or the occurrence and onset of frost for agriculture production [37], these biases could be significant enough to prohibit their use directly. However, given the relatively high correlation of daily values, for applications where only relative anomalies are required, such as for the calculation of trends and indices [38], CHIRTS minimum temperatures may already be fit for purpose.

Although the CHIRTS minimum temperature biases are larger, the lower standard deviation of errors and lower RMSE values suggest the biases are more systematic than from ERA5. This is a promising result for the use of CHIRTS minimum temperature data where the systematic biases could be well-estimated and corrected for, such as by infilling or extending an existing station record, or where a nearby station record is available for calibration. Since correlations of daily values are already high, bias-adjusted CHIRTS minimum temperature values could also be fit for purpose where direct values are required. A seasonally dependent bias correction could also be considered where the bias varies seasonally.

CHIRTS minimum temperatures perform worse at Kisumu and Saltpond, which both border water bodies. Performance of CHIRTS may be dependent on a number of location specific factors and the complexity of the topography; however, this would need further investigation in a study at more locations.

The results for maximum temperatures are good across all measures and show an improved performance of CHIRTS compared to ERA5 and ERA5-Land. ERA5 and ERA5-Land underestimate maximum temperatures by between 1 °C and 4.4 °C at all but one location, whereas the CHIRTS daily bias was between −0.5 °C and 0.5 °C at all locations except Kisumu.

On an annual basis, CHIRTS generally has higher correlations of mean maximum temperature values than ERA5 and ERA5-Land. CHIRTS exhibits statistically significant trends of similar magnitudes to those from the station data more consistently than ERA5 or ERA5-Land.

CHIRTS also estimates extreme maximum temperatures well. It could be expected that gridded data exhibits lower extreme values than point-based measurements, given their areal nature, which naturally leads to the smoothing of extreme values. However, the CHIRTS annual extreme maximum temperatures are comparable to those of the station data across the locations. This is even true at Sadore, Niger, which has the highest extreme temperatures often exceeding 45 °C, showing the ability of CHIRTS to accurately estimate extremely high temperatures, which are often critical to many studies.

The fact that CHIRTS maximum temperatures show strong performance when comparing daily values, annual means and extremes is very promising for the use of CHIRTS maximum temperatures in a wide variety of applications, both as absolute and relative values.

The results of this study show the CHIRTS dataset to be a promising addition to the set of gridded data products that provide near-global, long-term, high-resolution air temperature estimates. CHIRTS data are based on satellite data and incorporate station data records; hence, they complement and potentially add additional information to existing reanalysis temperature data products when used in combination or alone. Under many measures, CHIRTS performed similar or better than ERA5 and ERA5-Land records in comparison with data from eight stations across Africa. CHIRTS data are also on a resolution higher than existing temperature products, which could support better understanding of local climates. Although the number of locations in this study is relatively small, it shows promis-

ing findings of the accuracy of CHIRTS daily minimum and maximum temperature records across a diverse set of climates in Africa. Further studies focusing on specific geographic areas or certain terrains, such as carried out by Reda, Liu, Tang and Gebremicael [34] would complement this study and could provide more detailed results about the performance of CHIRTS under specific conditions, particularly in complex terrains.

**Author Contributions:** D.P. conceptualized and designed the research with input from D.S., D.N. and M.B.S. D.P. performed the analysis and prepared the manuscript. D.S., D.N. and M.B.S. provided critical comments on previous versions of the manuscript and input ideas and feedback following discussions. All authors have read and agreed to the published version of the manuscript.

**Funding:** This research was funded by a grant from the African Institute for Mathematical Sciences, www.nexteinstein.org, with financial support from the Government of Canada, provided through Global Affairs Canada, www.international.gc.ca, and the International Development Research Centre, www.idrc.ca (accessed on 26 June 2022).

**Data Availability Statement:** Satellite and reanalysis data used in the study are freely available online from the corresponding data sources cited in the article. Station data used in this study from Ghana, Kenya, Tanzania and Zambia are available from the respective national meteorological services of those countries, but restrictions may apply to the availability of these data because of the policies of the data owners. Data are available from the corresponding authors upon reasonable request and with permission of the respective national meteorological services. The station records from Sadore, Niger were provided by the International Crops Research Institute for the Semi-Arid Tropics (ICRISAT) and are available from the corresponding author upon reasonable request.

**Acknowledgments:** The authors thank and appreciate the Climate Hazards Center, UC Santa Barbra, and the European Centre for Medium-Range Weather Forecasts (ECMWF) for making satellite and reanalysis temperature data available, and the national meteorological services of Kenya, Ghana, Tanzania and Zambia, and the International Crops Research Institute for the Semi-Arid Tropics (ICRISAT) for providing station temperature data.

**Conflicts of Interest:** The authors declare no conflict of interest.

**Code Availability:** Code for data cleaning and analysis associated with this paper is available at https://github.com/dannyparsons/chirts-paper (accessed on 26 June 2022).

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
