# Peer review of "Evaluation of Satellite-Based Air Temperature Estimates at Eight Diverse Sites in Africa"

_climate, doi:10.3390/cli10070098_

Round 1

Reviewer 1 Report

The authors evaluated the CHIRTS-daily compared to measurements from eight ground stations throughout Africa. The authors also benchmarked the CHIRTS performance against the ERA5 and ERA5-Land reanalysis to understand its potential to provide localized temperature information.

The research topic is relevant and has applications in several areas of human activities. The methodology is consistent with the goals. The authors provided a detailed description of the climate regime in-ground sites and statistical indexes. I only recommend the authors provide details on the quality check procedure and criteria. 

Finally, I suggest the authors use the Taylor diagram for performance comparison between ERA-5, ERA-5 Land, and CHIRTS. The Taylor diagram allows a better understanding of the performance difference between the three databases based on a visual tool of the statistical indexes. The authors can find a good summary of the Taylor diagram at https://www.ncl.ucar.edu/Applications/taylor.shtml.

Author Response

We thank the reviewer for their review and constructive feedback on the manuscript.

Point 1: I only recommend the authors provide details on the quality check procedure and criteria. 

Response 1: Details of the quality control tests have been added to lines 122-129, including a reference to the WMO guidelines on Station Data Quality Control where these tests were derived from. We appreciate this suggestion to include these details.

Point 2: Finally, I suggest the authors use the Taylor diagram for performance comparison between ERA-5, ERA-5 Land, and CHIRTS. The Taylor diagram allows a better understanding of the performance difference between the three databases based on a visual tool of the statistical indexes. The authors can find a good summary of the Taylor diagram at https://www.ncl.ucar.edu/Applications/taylor.shtml.

Response 2: We thank the reviewer for this suggestion. After carrying out some further analysis based on this suggestion we found that the Taylor Diagram provided a very good visual summary of what we wanted to communicate about the performance of the minimum temperatures of CHIRTS vs ERA5 vs ERA5-Land, and it complemented the other results. We have therefore included Taylor Diagrams (split by station) in lines 320-324 and referenced it in the paragraph above to explain its interpretation.

Point 3: (Comment on line 203) I suggest to join the last phrase in the earlier paragraph.

Response 3: This paragraph has been expanded in lines 214-221 in response to comments from another reviewer. Therefore, there is now no longer a single sentence paragraph in this position.

Point 4: (Various comments throughout) please, verify the Latex code.

Response 4: Please note that the “Error! Reference source not found” message only seems to appear in the PDF version of the paper sent to the reviewers and does not appear in the Word version. The authors will confirm with the editor that this issue is resolved.

Reviewer 2 Report

The objective of this study is to evaluate the performance of CHIRTS-daily min and max temperature records in Africa through comparison with station data records. This paper is very well written and the figures are appreciate. I only have some minor questions. 

1. Please check the citations. It appears as "Error! Reference source not found" in the text. 

2. Line 123, can you give a detailed discussion on what has been done for data qaulity control?

3. Line 204, I am curious why we see differences in the seasonal patterns. Can you give some explanations? 

4. Figure 3 and 4, it is interesting how the correlation is impacted by precipitation amount. Can you explain why we have extremely low T correlation when higher precipitation is observed? 

Author Response

We thank the reviewer for their review and constructive feedback on the manuscript.

Point 1: Please check the citations. It appears as "Error! Reference source not found" in the text.

Response 1: Please note that the “Error! Reference source not found” message only seems to appear in the PDF version of the paper sent to the reviewers and does not appear in the Word version. The authors will confirm with the editor that this issue is resolved.

Point 2: Line 123, can you give a detailed discussion on what has been done for data quality control?

Response 2: Details of the quality control tests have been added to lines 122-129, including a reference to the WMO guidelines on Station Data Quality Control where these tests were derived from. We appreciate this suggestion to include these details.

Point 3: Line 204, I am curious why we see differences in the seasonal patterns. Can you give some explanations?

Response 3: Details explaining the differences in the seasonal pattern observed across different areas of Africa has been added to lines 214-221, which relate to rainfall patterns, proximity to the equator and northern/southern hemisphere difference, has been added. We appreciate the suggestion of this addition to explain this to the reader.

Point 4: Figure 3 and 4, it is interesting how the correlation is impacted by precipitation amount. Can you explain why we have extremely low T correlation when higher precipitation is observed?

Response 4: This is a useful point to draw out and clarify. Lower correlation does appear to coincide with the rainy periods. However, the rainy periods also coincided with reduced temperature variability, and this can adversely affect the correlation as there is a smaller range to the data. We have attempted to clarify this relationship in the paragraph in lines 259-274, which also explains that RMSE is not reduced in rainy periods, indicating that the reduced correlation is likely caused by the reduced variability, and not an indication of worse performance in terms of deviations from the station values. This idea is also represented visually in Figure 6 for one station. We appreciate the opportunity to clarify this important point for readers.

Reviewer 3 Report

The manuscript entitled “Evaluation of satellite-based air temperature estimates at eight diverse sites in Africa” (climate-1774337) says about the complement of the sparse networks of surface air temperature measurements on ground meteorological stations in Africa with high resolution temperature data from satellite and reanalysis. This manuscript could be interesting to readers because it deals with the important issue of finding new data on the atmosphere (here only on air temperature) beyond the classical instrumental observation of the atmosphere. However, the manuscript has disadvantages, which should be corrected in order to become a scientific paper. It will be mentioned here only one key disadvantage (DK) in the manuscript and several other disadvantages (DO), which should guide the authors in preparing a new revised version of the manuscript.

DK1.) The manuscript processes, formally compare and complement data of very high spatial resolution of 0.05 degrees and data of very small spatial resolution of 20-30 degrees (this is a very rough estimate), such as 8 selected meteorological stations for the whole of Africa (Figure 1 and Table 1). There is a problem here and answers to the following questions are being sought. What is actually being improved and what has been gained? In the first case, when the resolution is very high, atmospheric processes of a wide range of scales are included, from the smallest scales to the largest scales. In the second case, only large-scale processes are observed, because small-scale processes are not seen in a rare network of observation stations. The paper does not see how the authors harmonized this and did they consider it at all? Therefore, I suggest that the authors write something about it, and above all, estimate the errors that occur as a result, etc. This is usually done by including spectral analysis and similar considerations.

DO1.) The following phantom text appears in several places in the manuscript: "Error! Reference source not found." That text must be removed.

DO2.) Lines 12-27: (Abstract): Abbreviations should not be used in the abstract without explanation, especially if they are not generally known.

DO3.) Lines 111-112: (Figure 1): Figure 1 does not show the contours of Africa. That needs to be fixed.

DO4) All mathematical variables used in the manuscript must have the same writing font throughout the manuscript. For example, in the case of RMSE value, this is not the case. RMSE is written somewhere in italic font, and somewhere in normal font.

Author Response

We thank the reviewer for their review and constructive feedback on the manuscript.

Point 1: DK1.) The manuscript processes, formally compare and complement data of very high spatial resolution of 0.05 degrees and data of very small spatial resolution of 20-30 degrees (this is a very rough estimate), such as 8 selected meteorological stations for the whole of Africa (Figure 1 and Table 1). There is a problem here and answers to the following questions are being sought. What is actually being improved and what has been gained? In the first case, when the resolution is very high, atmospheric processes of a wide range of scales are included, from the smallest scales to the largest scales. In the second case, only large-scale processes are observed, because small-scale processes are not seen in a rare network of observation stations. The paper does not see how the authors harmonized this and did they consider it at all? Therefore, I suggest that the authors write something about it, and above all, estimate the errors that occur as a result, etc. This is usually done by including spectral analysis and similar considerations.

Response 1: We thank the reviewer for raising this point and for the opportunity to clarify these methodology points. Firstly, the eight station are not intended to be representative of the entire continent and are not intended to represent large scale areas of 20-30 degrees. A much larger sample of stations would of course be required if this was the aim.

Rather, the aim is to provide an indication of the performance of CHIRTS at a small but diverse set of individual locations across Africa, but not to represent the entire area of the continent. This point is mentioned in the abstract and we have clarified this intention in the aim of the study in line 80 to avoid potential confused to the reader. The point is reenforced in the Conclusions, where we mention that further studies focusing on smaller areas and utilising denser networks of stations would complement the results of this study.

We would like to clarify that this study does not include a spatial analysis of the gridded data. Only the individual pixels located closest to the stations are used for comparison, and not that of a larger area. Therefore, the results are not intended to be representative of large areas or the entire continent, but are instead point-to-pixel comparisons of the gridded data to indicate performance at specific locations. We have added further clarification of the comparison method used in Section 2.3 (line 80) and have emphasised the point on specific locations rather than large scale areas in Section 2.4 (lines 163-165), which we hope provides further clarity on this for the reader. With this as the intention of the paper, we believe that concept of a need for estimation of any possible large scale vs small scale spatial errors mentioned is not relevant for this study, with its stated aims.

Point 2: DO1.) The following phantom text appears in several places in the manuscript: "Error! Reference source not found." That text must be removed.

Response 2: Please note that the “Error! Reference source not found” message only seems to appear in the PDF version of the paper sent to the reviewers and does not appear in the Word version. The authors will confirm with the editor that this issue is resolved.

Point 3: DO2.) Lines 12-27: (Abstract): Abbreviations should not be used in the abstract without explanation, especially if they are not generally known.

Response 3: We believe the reviewer may be referring to the term “ERA5” in the abstract. Although ERA5 is typically written in capitals, the authors are not aware of this being as an acronym of a longer term. See the paper Hersbach et al. (2020) which introduced the ERA5 dataset and simply refers to it as “the ERA5 reanalysis” https://doi.org/10.1002/qj.3803. We are happy to include a suggested further explanation in the abstract if this is requested.

Point 4: DO3.) Lines 111-112: (Figure 1): Figure 1 does not show the contours of Africa. That needs to be fixed.

Response 4: While some maps do include contours, the authors felt that we did not want to draw the readers attention to the spatial features of the map, but rather focus on the locations of the individual stations within the continent. This relates to Response 1 where the authors did not want to imply that the locations are fully representative of the topography of the entire continent, or that results at individual locations would represent performance at large scales. Therefore, the authors decided not to include contours on this map.

Point 5: DO4) All mathematical variables used in the manuscript must have the same writing font throughout the manuscript. For example, in the case of RMSE value, this is not the case. RMSE is written somewhere in italic font, and somewhere in normal font.

Response 5: We appreciate this inconsistency being brought to our attention and will confirm with the editor that this issue is resolved.

Round 2

Reviewer 3 Report

The manuscript entitled “Evaluation of satellite-based air temperature estimates at eight diverse sites in Africa” (climate-1774337) has been significantly improved over the previous version and it is recommended for publication.